# Photocured room temperature phosphorescent materials from lignosulfonate

Hongda Guo[1], Mengnan Cao[1], Ruixia Liu[1], Bing Tian [1], Shouxin Liu[1], Jian Li[1], Shujun Li [1], Bernd Strehmel[2], Tony D. James [3,4] ✉ & Zhijun Chen [1] ✉

Photocured room temperature phosphorescent (RTP) materials hold great potential for practical applications but are scarcely reported. Here, we develop photocured RTP materials (P-Lig) using a combination of lignosulfonate, acrylamide, and ionic liquid (1-ethyl-3-methylimidazolium bromide). With this design, lignosulfonate simultaneously serves as RTP chromophore and photoinitiator. Specifically, lignosulfonate in the ionic liquid generates radicals to polymerize the acrylamide upon UV irradiation. The resulting lignosulfonate is automatically confined in an as-formed crosslinked matrix to provide RTP. As such RTP with an emission lifetime of ~110 ms is observed from the confined lignosulfonate in P-Lig. Additionally, energy transfer occur between P-Lig and Rhodamine B (RhB), triggering red afterglow emission when P-Lig is in situ loaded with RhB (P-Lig/RhB). As a demonstration of potential applications, the P-Lig and P-Lig/RhB are used as photocured RTP coatings and RTP inks for fabricating 3D materials and for information encryption.

Room temperature phosphorescent materials can be used for a wide range of applications, including organic light-emitting diodes (OLEDs), scintillators, and anti-counterfeit applications[1-4]. Amongst the available RTP materials, organic RTP materials have attracted particularly attention since they are flexible, exhibit tunable optical properties, require mild preparation and facilitate easy processing[5-8]. In general, two crucial design principles must be followed to ensure that such RTP materials are obtained: (a) intersystem crossing (ISC) from the lowest excited singlet state ($S_1$) to a triplet state ($T_n$) and (b) radiative transitions from the lowest excited triplet state ($T_1$) to the ground state ($S_0$)[9]. Following these rules, organic RTP materials, including, supramolecular systems, molecular crystals, MOF, polymer composites and carbon dots have been developed[10-16].

To realize such RTP materials for use in the as-mentioned applications, these RTP materials are firstly dispersed in aqueous or organic solvent. After that, the dispersion is coated onto the surface of a substrate and then subjected to physical drying by either heating or natural evaporation. Significantly, this process appears quite time-consuming. In addition, the solvent drying process might alter the matrix used, which is undesirable. This motivated us to develop an approach to decorate RTP materials using a photocuring process, which was conducted at ambient room temperature over short periods of time using non-contact light sources[17,18]. However, such photocured RTP materials have rarely been reported[19]. In addition, the photocured RTP formula requires a photosensitizer and RTP chromophores, increasing the complexity and stability of the whole system. Moreover, the photosensitizer could also quench the RTP emission from the chromophores.

Lignin is the most abundant aromatic biomass resource in nature and exhibits interesting optical properties, such as UV blocking, fluorescence, photothermal conversion, RTP, and photocatalytic properties[20-32]. In this work, we employ lignosulfonate to serve as RTP chromophore and photoinitiator simultaneously. Specifically, lignosulfonate generates radicals to polymerize the acrylamide in the

[1]Key Laboratory of Bio-based Material Science & Technology, Northeast Forestry University, Ministry of Education, Harbin 150040, China. [2]Department of Chemistry, Institute for Coatings and Surface Chemistry, Niederrhein University of Applied Sciences, Adlerstr. 1, D-47798 Krefeld, Germany. [3]Department of Chemistry, University of Bath, BA2 7AY Bath, UK. [4]School of Chemistry and Chemical Engineering, Henan Normal University, Xinxiang 453007, P. R. China. ✉e-mail: T.D.James@bath.ac.uk; chenzhijun@nefu.edu.cn

ionic liquid (1-ethyl-3-methylimidazolium bromide) upon UV irradiation. As a result, the lignosulfonate is confined in the as-formed bulk and transparent matrix (P-Lig) generating a material able to generate RTP from the confined lignosulfonate in P-Lig. Significantly, this design can efficiently prevent the quenching of RTP that occurs when using a conventional formula consisting of standard photoinitiators and RTP chromophores. Figure 1 outlines a schematic description of the process.

## Results

### Photocured properties and RTP performance

2D heteronuclear single quantum coherence (HSQC) NMR spectra reveals that lignosulfonate contains aromatic units decorated by -OMe moieties (Supplementary Fig. 1). Such a structure enables lignosulfonate to exhibit absorbance over a UV–Vis range. As expected, lignosulfonate exhibited an absorbance in the range of 300–400 nm in the UV–Vis spectra, indicating lignosulfonate could absorb and use UV light for the following steps (Supplementary Fig. 2). After being exposed to the UV light (365 nm), photochemical radicals were formed including quinone and hydroxyl radicals were generated by lignosulfonate as determined by ESR spectra (Fig. 2a and Supplementary Fig. 3)[26,33,34]. Theoretically, concentrated lignosulfonate can generate more photoradicals. However, concentrated lignosulfonate also results in more phenolic moieties, which can reduce the amount of photoradicals. Thus, it is important to seek a suitable concentration of lignosulfonate for the maximum generation of photoradicals. From ESR spectral analysis we determined the best concentration range of lignosulfonate for generating photoradicals as 0.1–0.15% w/w in the ionic liquid (Supplementary Fig. 4). As such we anticipated that the radicals generated by lignosulfonate upon UV irradiation, could be used to initiate free radical polymerization (Fig. 2a). Therefore, a mixture of lignosulfonate, ionic liquid and acrylamide was exposed to UV light for photocuring. The photocuring process was monitored using FT-IR and the double bond conversion reached ~96% upon UV exposure for 20 min (Fig. 2b and Supplementary Fig. 5). In addition, the effect of lignosulfonate on the polymerization process was evaluated. Using the ESR results as guide a concentration of lignosulfonate from 0.1% to 0.15% w/w was trialed and it was determined that concentrated lignosulfonate induced a faster rate of polymerization (Supplementary Fig. 6). To prove the light-mediated formation of radicals in the curing process, 5,5-dimethyl-1-pyrroline N-oxide (DMPO), as a radical scavenger, was added to the photocuring formula. As expected, no polymerization was observed upon UV irradiation for 20 min (Supplementary Fig. 7). Interestingly, lignosulfonate does not cure the formula upon UV irradiation in the aqueous solution and only ~10% double bond conversion was achieved. This is because lignosulfonate in ionic liquid generates more reactive radicals as determined by ESR spectra (Supplementary Fig. 8). [1]H NMR and in situ FT-IR analysis indicated that lignosulfonate was covalently attached to the polyacrylamide through the reaction between the phenolic group and the

double bond and a new C–O–C bond was formed in the reaction. (Supplementary Fig. 9)[35,36].

Attributed to the inherent priorities of photocured materials, P-Lig with different shapes were easily obtained by in situ exposure of the precursors in a template to UV sources (Fig. 2c). In addition, the photocured P-Lig exhibited high transparency over the visible range (Supplementary Fig. 10). Interestingly, the precursors of P-Lig displayed no RTP emission before photocuring. Therefore, the liquid surrounding of P-Lig contributes to RTP quenching. To identify the reason for the RTP quenching, lignosulfonate was dissolved in ionic liquid without acrylamide and no RTP emission was observed (Supplementary Fig. 11). Thus, the free polymer chains of lignosulfonate in organic solvents facilitate return of the excited state to the ground state due to enhanced nonradiative relaxation, resulting in RTP quenching. However, P-Lig exhibited strong RTP emission at 510 nm after photocuring (Fig. 2d). The phosphorescence quantum yield of P-Lig is 11.04%. More interestingly, both the RTP intensity and lifetime was enhanced with an increased duration of photocuring (Fig. 2e and Supplementary Fig. 12) caused by a decrease in diffusion processes with prolonged conversion. As such, the lifetime increased to ~110 ms after photocuring for 20 min. To verify the reproducibility of the process, it was repeated 5 times and the as-obtained materials exhibited similar lifetime of ~110 ms (Supplementary Fig. 13). While the lifetime of P-Lig decreased to 24 ms upon humidity treatment (Supplementary Fig. 14) due to the good compatibility of polyacrylamide with water. As such, the relationship between the RTP of P-Lig and water was systematically investigated. The RTP lifetime continuously decreased after immersing P-Lig in water for extended times (Supplementary Fig. 15). After 30 min, the material exhibited a reduced lifetime of ~13.7 ms and after 40 min, no RTP emission could be detected. However, the lifetime recovered when the sample was dried at 80 °C for 60 min underlining again the findings disclosed *vide supra*. The lifetime was fully recovered after repeating such humidity-drying processes up to 5 times. However, immersing the P-Lig in different organic solvents, such as, N,N-dimethylformamide, dichloromethane, tetrahydrofuran, acetonitrile and ethanol for three months did not quench the RTP emission (Supplementary Fig. 16). Since the polyacrylamide possesses a low compatibility with these solvents, the solvents are unable to quench the RTP emissions. Quantitative analysis further indicated that the lifetime of P-Lig does not obviously change after treatment by organic solvents (Supplementary Fig. 17). The effect of temperature on P-Lig was also evaluated. The RTP emission and lifetime of P-Lig were measured at different temperatures and an increased temperature promoted the non-radiative migration of the excitons of lignosulfonate, which compromised both the RTP intensity and lifetime (Supplementary Fig. 18). To confirm the advantages of our formula, several control experiments were conducted. In these control samples, Ir2959 or benzophenone were used as photoinitiators with lignosulfonate as the RTP chromophore. Indeed, all these formulae were photocured after UV irradiation (Supplementary Fig. 19).

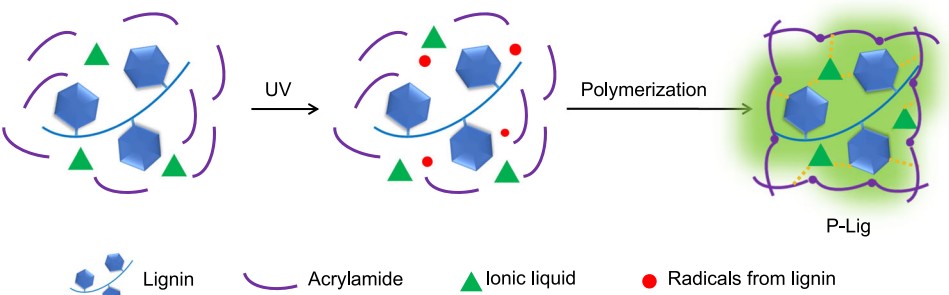

Lignin Acrylamide Ionic liquid Radicals from lignin

**Fig. 1 | Schematic illustration for the preparation of P-Lig using a photocuring process assisted by lignin in the presence of acrylamide and ionic liquid.**

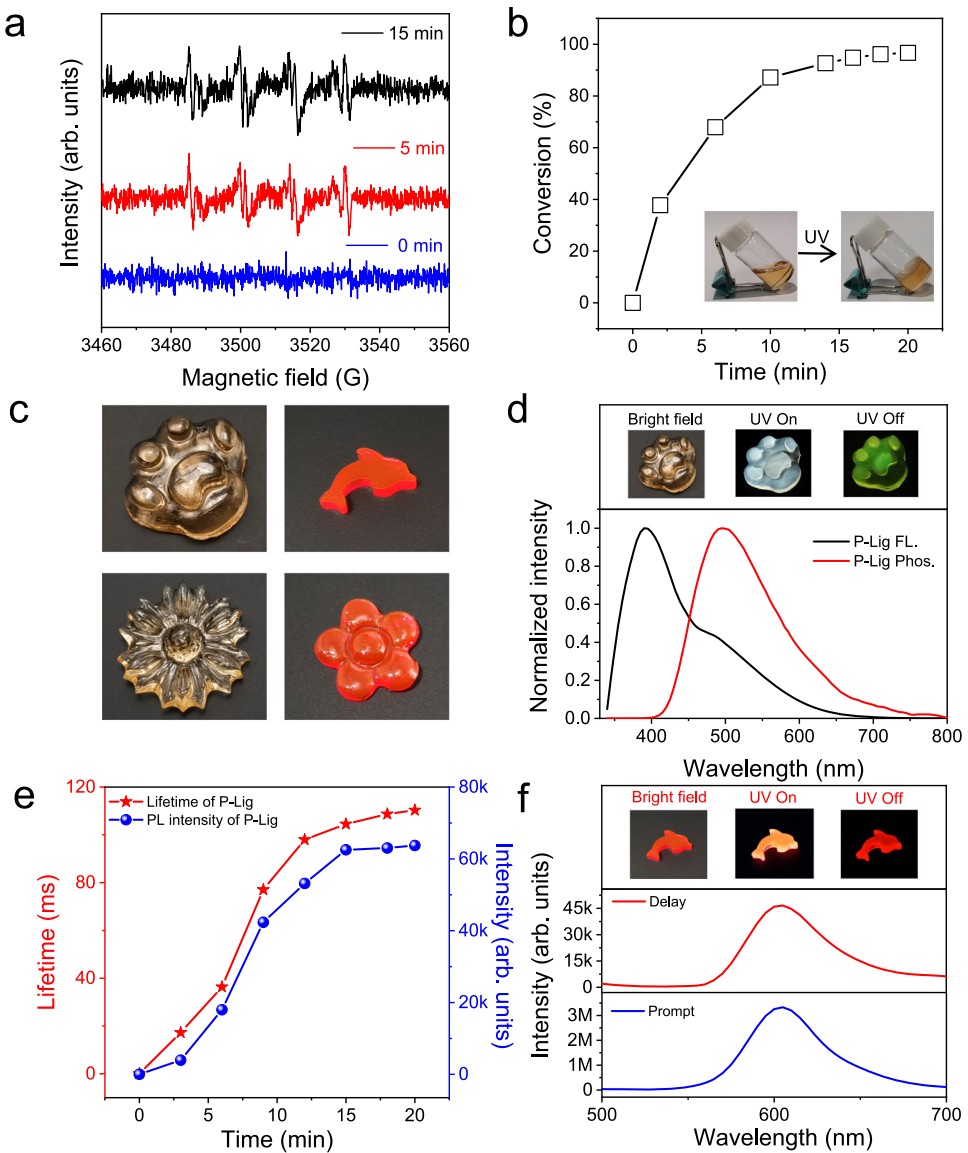

**Fig. 2 | Preparation and RTP emission of P-Lig and P-Lig/RhB. a** ESR spectra of lignosulfonate in ionic liquid upon UV irradiation for 0 min (blue line), 5 min (red line) and 15 min (black line) in the presence of DMPO; **b** Conversion of double bonds of P-Lig upon UV irradiation($\lambda_{exc.}$= 320 nm, $\lambda_{collected}$ = 510 nm, delay time = 10 ms); **c** Images of P-Lig (left) and P-Lig/RhB (right); **d** Standard (black line) and delayed (red line) emission spectra of P-Lig upon excitation with 320 nm light, Inset: the images of P-Lig in daylight (left), P-Lig upon excitation with a UV light source (middle) and P-Lig after switching off the UV light source (right), the delay time for the measurements was 10 ms; **e** RTP intensity (blue line) and lifetime (red line) of P-Lig obtained by different photocuring durations ($\lambda_{exc.}$= 320 nm, $\lambda_{collected}$ = 510 nm, delay time = 10 ms); **f** Standard (blue line) and delayed (red line) emission spectra of P-Lig/RhB upon excitation with 320 nm light, Inset: the images of P-Lig/RhB in daylight (left), P-Lig/RhB upon excitation with a UV light source (middle) and P-Lig/RhB after switching off the UV light source (right), the delay time for the measurements was 10 ms.

However, the lifetime of the as-obtained materials were 42.63 and 36.97 ms, which are much shorter than P-Lig. Then in order to demonstrate the general quenching effect of the photoinitiator on the chromophore, phenylboronic acid was used as a chromophore in a photocured RTP formula initiated by Ir2959 or benzophenone. The lifetime of the as-obtained samples was shorter than the cured samples initiated by ammonium persulphate (APS) (Supplementary Fig. 19). Thus, the optical properties of these RTP chromophores are negatively affected by an additional external photoinitiator. In addition, the cured samples initiated by lignosulfonate exhibited higher mechanical performance including hardness and young's modulus than the photocured resin using the RTP chromophore (phenylboronic acid) or without a RTP chromophore initiated by a typical photoinitiator (Ir2959) (Supplementary Fig. 20)[37]. The robust mechanical performance of RTP materials is particularly beneficial for their application as

coatings or functional components in devices. Significantly, the formula could be varied to include other monomers. A series of monomers including acrylic acid, methyl acrylate, methyl methacrylate, styrene and N-isopropylacrylamide were successfully used to produce P-Lig initiated by lignosulfonate in ionic liquid. All the samples can be photocured in 20 mins and exhibited RTP emission with lifetimes of 76.96, 62.95, 57.95, 67.19 and 85.56 ms (Supplementary Fig. 21). In addition, the source of lignin could be varied, kraft lignin, alkali lignin and enzymatic hydrolysis lignin could also serve as photoinitiators for the reaction. All the samples were photocured in 20 mins with double bond conversions of 89.27%, 97.24% and 73.89%, respectively. In addition, all of these cured samples generate RTP emission (Supplementary Fig. 22).

Markedly, the absorbance of RhB partially overlaps with the RTP emission spectra of P-Lig, indicating that energy transfer could occur

between RhB and P-Lig (Supplementary Fig. 23)[38]. Therefore, the optical behavior of P-Lig/RhB was investigated. P-Lig/RhB generated a fluorescence emission at ~600 nm, attributed to the RhB. While P-Lig/RhB also generated delayed RTP emission at ~600 nm with a lifetime of ~39 ms (Fig. 2f). The fluorescence and delayed fluorescence quantum yield of P-Lig/RhB is 57.62%. These results indicate that energy transfer occurs between P-Lig and RhB in P-Lig/RhB, triggering the red afterglow emission of RhB. To further understand the energy transfer process, the donor (P-Lig) quenching data, including donor emission intensity, and corresponding lifetime, with varying concentrations of RhB added to the system were determined (Supplementary Fig. 24). Significantly, no delayed emission for RhB was observed upon direct excitation at 550 nm, but a strong delayed emission was observed by indirect excitation at 320 nm (excitation of the donor) (Supplementary Fig. 24). All these results confirmed that triplet-to-singlet FRET occurs between lignosulfonate and RhB in the photocured matrix. The efficiency was 16.5%, 19.4%, 23.9%, 34.0%, 45.9%, 64.6%, and 79.7% with different concentrations: 0.01%, 0.03%, 0.05%, 0.07%, 0.08%, 0.09% and 0.1% of RhB (Supplementary Table 1)[38–40].

## Mechanism

To further understand the role of lignosulfonate in the RTP emission, a control formula without lignosulfonate and consisting of acrylamide and 1-ethyl-3-methylimidazolium bromide and initiated by ammonium persulfate was prepared. The double bond conversion was 99% as determined by FT-IR spectra. However, the as-prepared sample only exhibited weak and short-lived RTP emissions,

attributed to clustering-induced emission of the carbonyl moieties in as-formed polymer chains (Supplementary Figs. 25 and 26)[41,42]. This result confirmed that lignosulfonate was the main chromophore for the RTP emission. Meanwhile, in situ FT-IR spectra indicated that the carbonyl signals of the formula shifted from 1677 to 1670 $cm^{-1}$, suggesting hydrogen bonding interactions were enhanced during the photocuring process (Fig. 3a)[43]. In addition, theoretical simulations confirmed that polymerization of acrylamide increased binding interactions with the lignosulfonate molecules (Fig. 3b). Specifically, the binding energy between the acrylamide monomer was −9.29 eV and the value increased to −11.93 eV when 2 units of acrylamide were polymerized. Further polymerization of acrylamide (6 units) exhibited enhanced binding interactions with the lignosulfonate (−17.14 eV). As a comparison, the interaction between lignosulfonate and six monomers was only −13.82 eV (Supplementary Fig. 27). Such enhanced interactions can restrict molecular vibrations and as such are beneficial for RTP emission[44]. In addition, to further understand the performance of P-Lig, ionic structures with different anions (Br−, Cl− and $CH_3COO^-$) were used. All systems were photocured with a similar conversion yield of double bonds when they were exposed to a UV light source in the presence of lignosulfonate (Supplementary Fig. 28). However, both the RTP intensity and lifetime of the sample made from an ionic structure including the $CH_3COO^-$ anion were much lower than samples generated using Cl− or Br− ions. The results indicate that the external heavy-atom effect was crucial for effective RTP emission (Fig. 3c). All these results confirmed that the photocuring-enhanced molecular interaction and external heavy

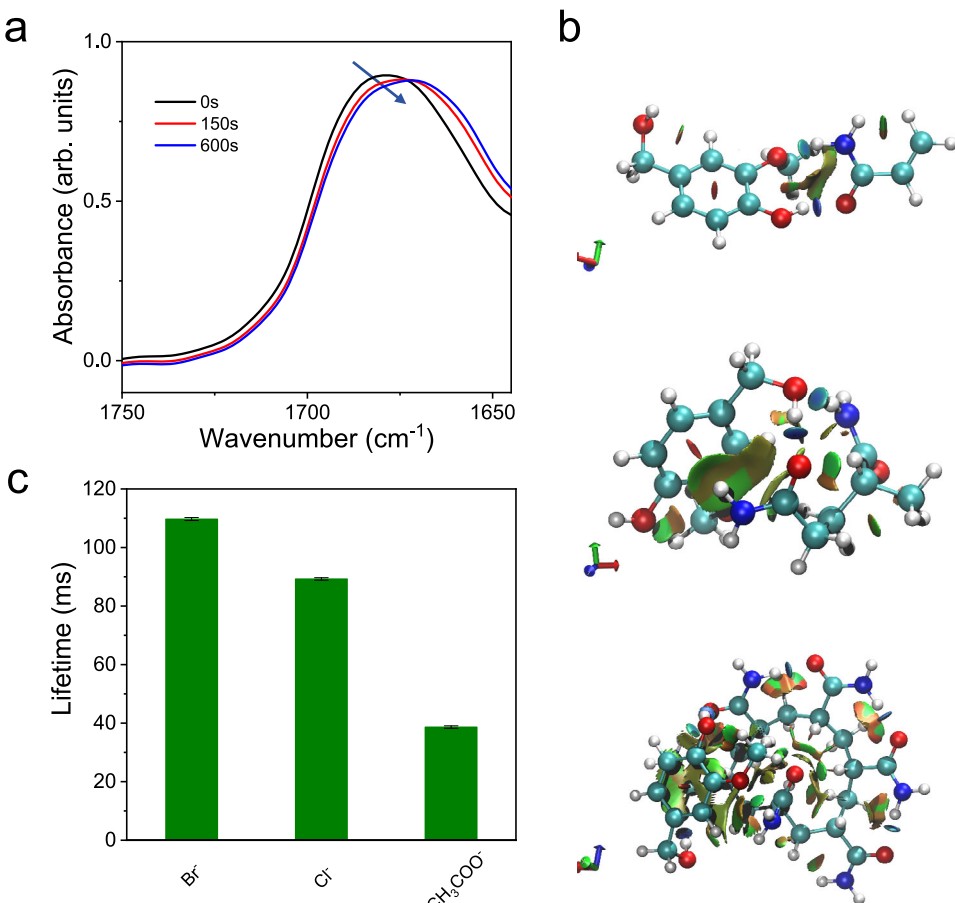

**Fig. 3 | Mechanism for the RTP of P-Lig. a** In situ FT-IR spectra of P-Lig upon UV irradiation for 0 s (black line), 150 s (red line) and 600 s (blue line); **b** Calculated interaction model between lignosulfonate and acrylamide with different polymerization degrees; **c** Lifetime of P-Lig prepared from ionic liquid with different anions (Error bars indicate the standard deviation for three separate measurements of the samples, the lifetime of the sample containing Br− is 109.72 ± 0.52 ms, the lifetime of the sample containing Cl− is 89.27 ± 0.43 ms, the lifetime of the sample containing $CH_3COO^-$ is 38.67 ± 0.41 ms).

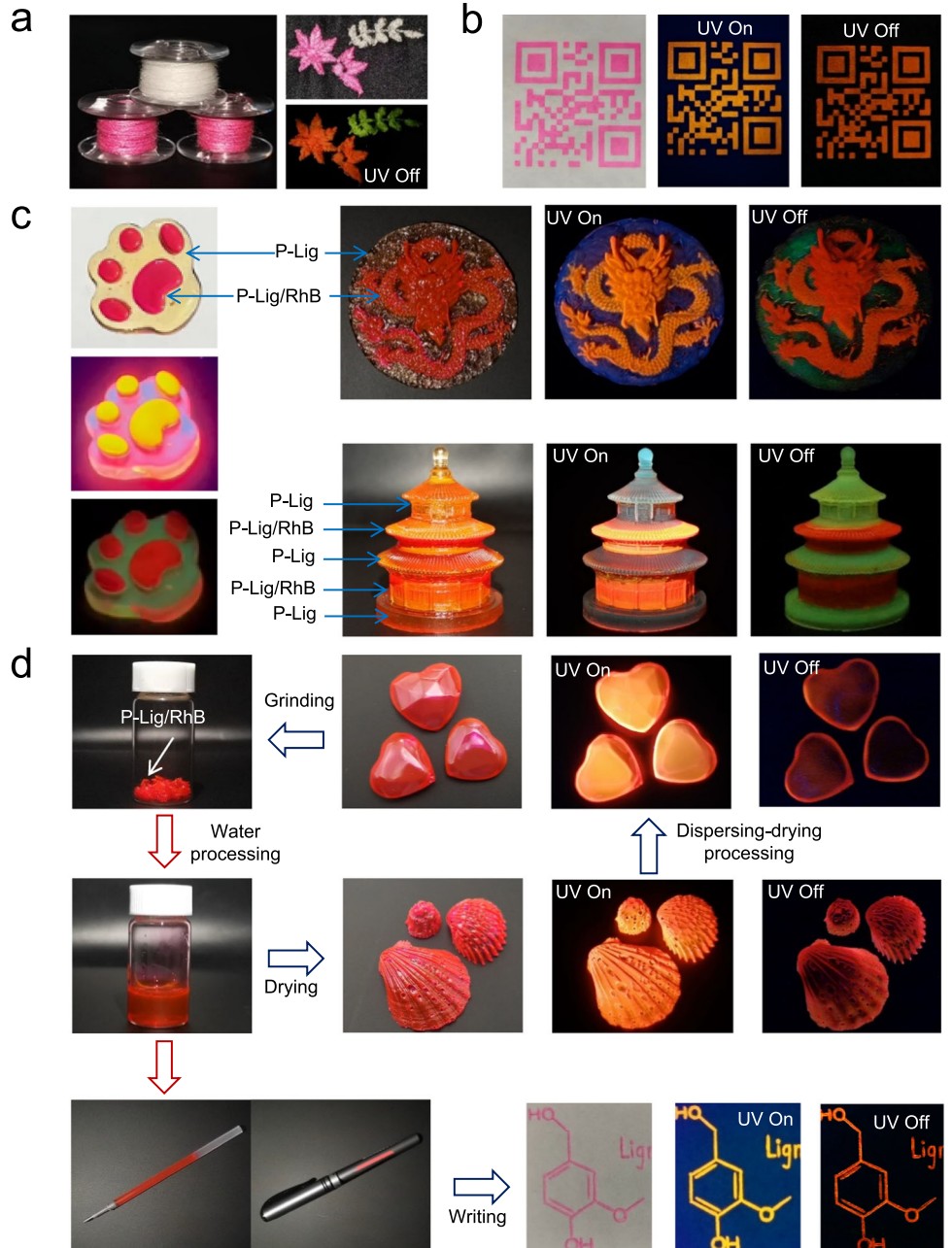

**Fig. 4 | Application of P-Lig and P-Lig/RhB. a** Images of embroidery made from yarns treated by P-Lig and P-Lig/RhB; **b** As-printed 2D Code on paper using P-Lig/RhB: Images in the bright field (left), images in the UV field (middle) and images after removing the UV light source (right); **c** The images of 3D bulk materials made from P-Lig and P-Lig/RhB in the bright field, UV field and after removing the UV light source; **d** Reprocessing 3D bulk materials made from P-Lig/RhB into different shapes and afterglow inks.

atoms in the system contribute to the effective RTP emission of lignosulfonate.

## Application

P-Lig and P-Lig/RhB were used to photo-cure coatings for cotton yarns (Fig. 4a). Specifically, the cotton yarns were immersed into formula containing solutions. After that, the treated yarns were exposed to UV irradiation. The coated yarns exhibited either green or red afterglow emission when treated by P-Lig or P-Lig/RhB, respectively. Encouraged by this, these RTP yarns were used to generate embroidery for anti-counterfeiting purposes using a commercial machine. Upon excitation using a 365 nm UV lamp, blue and red patterns of leaves from P-Lig yarns and P-Lig/RhB yarns, respectively, became clearly visible. When

the lamp was turned off, the portions of the embroidery made using P-Lig/RhB yarns maintained red emission. While, P-Lig yarns retained green afterglow emission. These observations clearly indicate that our strategy for preparing afterglow RTP yarns was successful and that they could be used in large-scale practical applications. To further demonstrate the potential, P-Lig/RhB was employed as printable RTP coatings. P-Lig/RhB was successfully printed onto a wide range of substrates, such as plastic, glass, paper, and PVA via screen printing methods (Fig. 4b and Supplementary Fig. 29). All these printed patterns exhibited afterglow RTP emission after removing the excitation sources. To demonstrate the advantages, a comparison between reported lignosulfonate RTP materials and P-Lig was conducted[22]. The tensile strength of the yarns (paper) decreased after the treatment with

previously reported lignosulfonate RTP materials (Supplementary Fig. 30). While, an obvious enhancement on mechanical performance was observed for the treated samples using P-Lig, attributed to the robust mechanical performance of the photocured P-Lig. In addition, P-Lig or P-Lig/RhB can be used as photocured RTP inks and for preparing 3D bulk materials. Several complicated 3D structures with RTP emission were obtained using a layer-by-layer photocuring method (Fig. 4c and Supplementary Movie 1). Interestingly, the as-formed 3D materials from P-Lig can be processed into different shapes assisted by water (Fig. 4d). Specifically, the bulk 3D materials were firstly immersed into water. After that, the as-obtained sticky solution can be further processed into a specific shape assisted by a template via evaporation-induced drying. The whole process was reversible. P-Lig/ RhB also exhibited potential as a portable solid ink (Fig. 4d). Specifically, red inks for use by a pen were obtained by dispersion of solid P-Lig/RhB into water. The information written by the as-obtained inks exhibited a red color in the bright field and UV field. While, afterglow emission was observed after removing the UV light source. This confirmed that the ink could be used for information encryption and anti-counterfeiting applications. In addition, ~96% of the ionic liquid used to prepare P-Lig was recyclable. $^1$H NMR analysis confirmed that the as-recycled ionic liquid displayed the same signals as virgin ionic liquid (Supplementary Fig. 31). Moreover, to demonstrate the activity of recycled ionic liquid, P-Lig was prepared using the recycled ionic liquid and exhibited RTP lifetimes of 110 ms, which are similar to P-Lig samples prepared using fresh ionic liquid (Supplementary Fig. 32).

## Discussion
In summary, we have developed a photocured RTP material with a lifetime of ~110 ms using lignosulfonate. Where lignosulfonate simultaneously served as both the photoinitiator and RTP chromophore in the as-formed material, greatly increasing the convenience of the whole process. In addition, red afterglow emission could be obtained by loading P-Lig with RhB via an energy transfer strategy. Attributed to the photocured material properties, the as-developed P-Lig and P-Lig/ RhB exhibited potential in many areas including the preparation of luminescent yarns, printing, 3D manufacturing, and functional inks. Considering the easy preparation, low cost and good performance, such materials could be prepared on a large scale and used for practical applications. Significantly, lignin is not limited to this preparation and can also be integrated with sustainable deep eutectic solvents and monomers for generating additional sustainable and high-performance RTP materials.

## Methods
### Photocuring of P-Lig
A mixture of acrylamide (500 mg, 7.03 mmol), lignosulfonate (1 mg), and 1-ethyl-3-methylimidazolium bromide (400 mg, 2.09 mmol) were heated at 80 °C until the mixture changed to a homogeneous solution. After that, the solution was exposed to UV LED light sources (365 nm, 170 mW cm$^{-2}$) for 20 min for photocuring.

### Control sample prepared by ammonium persulfate
A mixture of acrylamide (500 mg, 7.03 mmol), ammonium persulfate (1 mg), and 1-ethyl-3-methylimidazolium bromide (400 mg, 2.09 mmol) were heated at 60 °C for 30 min. After that, a cured control sample was obtained for measurement. The double bond conversion was 99% determined by FT-IR spectra.

### Photocuring of P-Lig/RhB
A mixture of acrylamide (500 mg, 7.03 mmol), lignosulfonate (1 mg), RhB (0.7 mg, 0.0015 mmol) and 1-ethyl-3-methylimidazolium bromide (400 mg, 2.09 mmol) were heated at 80 °C until the mixture changed to a homogeneous solution. After that, the solution was exposed to UV LED light sources (365 nm, 170 mW cm$^{-2}$) for 20 min for photocuring.

### Preparation of green afterglow yarns
Cotton yarns were immersed into a solution of precursor (500 mg acrylamide, 1 mg lignosulfonate and 400 mg 1-ethyl-3-methylimidazolium bromide). After that, the treated cotton fibers were exposed to UV LED (365 nm, 170 mW cm$^{-2}$) for 30 min to generate green afterglow yarns.

### Preparation of red afterglow yarns
Cotton yarns were immersed into the solution of precursor (500 mg acrylamide, 1 mg lignosulfonate, 400 mg 1-ethyl-3-methylimidazolium bromide and 0.7 mg RhB). After that, the treated cotton fibers were exposed to UV LED (365 nm, 170 mW cm$^{-2}$) for 30 min to give red afterglow yarns.

### Preparation of 3D bulk RTP materials
3D bulk materials with multicolor RTP emission were prepared using a layer-by-layer photocuring method. Pour a solution of P-Lig precursor into a silicone mold, a thin layers of P-Lig with green RTP emission was obtained after photocuring. Then a solution of P-Lig/RhB precursor was poured onto the photocured P-Lig layer to obtain a layer with red emission. In such manner, multilayered structured 3D materials with multicolor RTP emission can be obtained. For the dispersing-drying process, as-obtained 3D materials were dispersed into water and the dispersion was then injected into the template for producing the materials with a certain shape.

## Data availability
All relevant data are included in this article and its Supplementary Information files. Source data are provided with this paper. All data underlying this study are available from the corresponding author Zhijun Chen upon request. Source data are provided with this paper.

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

## Acknowledgements

Z.J.C. wishes to thank the National Natural Science Foundation of China (31890774) and Fundamental Research Funds for the Central Universities (2572022CG02). T.D.J. wishes to thank the University of Bath and the Open Research Fund of the School of Chemistry and Chemical Engineering, Henan Normal University (2020ZD01) for support.

## Author contributions

Conceptualization: Z.C., S. Li., and T.D.J.; Methodology: H.G. and B.S.; Investigation: H.G., M.C., and R.L.; Visualization: H.G., S. Liu., J.L., and S. Li.; Supervision: Z.C., S. Li., B.S., and T.D.J.; Writing-original draft: All authors; Writing-review and editing: All authors.

## Competing interests

The authors declare no competing interests.
