## [Peer Review File · Nature Communications]

Reviewers' Comments:

Reviewer #1:

Remarks to the Author:

Guo and co-workers presented a photocuring resin made from lignin, acrylamide and 1-ethyl-3-methylimidazolium bromide, which showed enhanced room temperature phosphorescence (RTP) as well as longer emission lifetime. The resin was further employed to trigger a red afterglow emission when using rhodamine B as an energy transfer acceptor. The authors demonstrated the potential applications of this resin as luminescent coatings and inks for information encryption. Although it is an interesting study, the authors have extensively reported the RTP properties of lignin and its derivatives (Nat Comm 2022 and 2023; CRPS 2021), and lignin has been reported as a photoinitiator for 3D printing (ACS Sus Chem Eng, 2020, 8, 10959). As a result, the simple combination of these two properties, in my opinion, did not generate sufficient new knowledge that could match the high standard of Nature Communications. The detailed comments are as follows:

1. In terms of the RTP coatings and inks for information encryption, what's the significant advantage of current design compared to other RPT systems? For example, RTP materials can be simply mixed with conventional photocuring resins. A photoinitiator may be needed, but the concentration is only around 1% or less.
2. The authors claimed that "the liquid surrounding of P-Lig contributes to RTP quenching" (line 92). Does it mean the free polymer chains of lignin in organic solvents make the excited state go back to ground state with more nonradiative relaxations, or the acrylamide has charge transfer interactions with lignin that can quench the emission?
3. Line 104, how to understand "Since the polyacrylamide possesses a low compatibility with these solvents, the solvents are unable to quench the RTP emissions"? Did the authors investigate the relationships between compatibility and RTP intensity?
4. What's the factor leading to short-lived RTP emission of the formula without lignin?
5. The lignin-acrylamide interaction model for theoretical calculations is based on one single lignin group and one acrylamide monomer? The proper comparison should be based on same amount of lignin groups as well as the monomeric units. For example, if the interaction between lignin and a 6-unit polyacrylamide oligomer is investigated, the comparable model for monomer should be lignin and 6 monomer molecules. The reality in photocuring resins is that numerous monomers exist surrounding the lignin.
6. What if other monomers are used in this system, such as acrylates, methacrylates, and styrenics? Does this strategy have potential broad applications in different occasions? A more systematic study with deep understanding would be more convincing.
7. The FTIR data may be provided for evaluating the double bond conversion.

Reviewer #2:

Remarks to the Author:

In the manuscript entitled "Title: Photocured room temperature phosphorescence materials from lignin" authors have developed a photocured phosphorescent material (P-Lig) using lignin, acrylamide and an ionic liquid. Authors have used the same material (lignin) that can act both as photoinitiator to initiate the polymerization of acrylamide upon UV irradiation, and as phosphor that is trapped inside the polymer network. Authors have also shown a red-afterglow by using rhodamine B as the acceptor in the energy-transfer process along with P-Lig. Finally, authors have also shown different applications using the same material as photo-curable coatings and inks for information encryption. Same author have shown phosphorescence using lignin (Nat. Commun. 13, 5508 (2022); Nat. Commun. 14, 2614 (2023)) in previous reports. Also, there are reports on photocurable phosphorescence (Adv. Optical Mater. 2300240 (2023)). Authors should elaborate more on the novelty aspects in the revised manuscript apart from addressing the following

concerns from my end.

1. Authors should mention the delay time used for delayed emission spectrum in Figure 2d.
2. In line no. 97, authors have mentioned the decrement of lifetime to 24 ms on humidity treatment. However, the lifetime plot for the same is not shown.
3. What is the phosphorescent quantum yields for these materials.
4. What is the type of energy-transfer happening in this case. FRET or TS-FRET. Does the triplet state of lignin is alone involved in the process. What is the efficiency of the energy-transfer process here. It is recommended to investigate further on the energy-transfer process happening here since the manuscript lacks convincing energy transfer data from P-Lig to RhB. To address this gap, provide gradual donor (P-Lig) quenching data, including donor emission intensity and corresponding lifetime, with varying concentrations of RhB in the system. Additionally, demonstrate whether, upon indirect excitation (exciting the donor), the emission intensity of RhB surpasses that of direct excitation. This analysis will provide crucial insights into the energy transfer dynamics between P-Lig and RhB.
5. It is advisable to clearly mention the excitation and collection wavelengths in the figure captions throughout the manuscript, including both the main text and supplementary information for better understanding for the readers
6. In line no. 131, authors have claimed that the "as-prepared sample only exhibited weak and short-lived RTP emissions". Please provide the corresponding steady-state emission spectra. Additionally, clarify whether this short-lived RTP emission is observed in the absence of lignin, and if so, provide insights into the source of this RTP emission.
7. Authors have achieved afterglow, and also utilized energy-transfer process to sensitize the delayed fluorescence from rhodamine B. In this context, the following papers should be cited: Angew. Chem. Int. Ed. 60, 19720 –19724 (2021); Adv. Funct. Mater. 30, 2003693 (2020).

Reviewer #3:

Remarks to the Author:

I would like to start by stating that the manuscript titled: "Photocured room temperature phosphorescence materials from lignin" reports a novel, easy-to-follow and "simple" approach to prepare RTP materials based on lignosulfonate. I think that this manuscript should be accepted for publication, however, I think that some modifications are necessary. I have divided my comments in different sections so it is easier for the authors to address.

Title:

The title of the manuscript gives the impression that it refers to a review article rather than a paper. I suggest the authors to change it and also replace the word "lignin" with "lignosulfonate"

Abstract, Introduction and Conclusions:

The abstract and conclusions sections contain the necessary information for a reader to get an overview of the paper and its most important findings. They are also well-written!

As for the introduction it is short and concise.

However, I suggest the word lignin to be replaced with lignosulfonate in the entire text as there are many lignin types that could fall into this category.

Results and Discussion:

- Why did the authors used lignosulfonates compared to other more industrially relevant technical lignins such as kraft? I think this should be included in the main manuscript.

Would the results be the same is the experiments were to be repeated with kraft lignin instead of lignosulfonate?

- Has the same behaviour, i.e., lignosulfonate act as radical generator, be reported elsewhere in the literature with a similar ionic liquid system? If yes, I think it should be mentioned somewhere in lines 79-81.

- It is well-known that lignin act as a radical scavenger due to its phenolic content. Can the

authors comment on this and how can they address the balance between radical generation and radical scavenging?

- Can the authors comment on how the temperature would affect the performance of the lignin-containing RTP materials?

Methods:

Can the authors comment on the reproducibility of the method they developed?

Also, was there any optimization performed for the quantities used in this formulation? Could perhaps a higher lignosulfonate content reduce the UV irradiation time?

General questions:

Can the authors comment on the recyclability of the ionic liquid used in their protocol? I think that this is quite important and it should be mentioned in the main manuscript too.

Based on the explanation in the main text of the manuscript as well as from the methods section, it is my understanding that the lignosulfonate content is kept as low as possible. However, since no purification is performed after the polymerization is complete, I am curious about the interaction of lignin with the rest of the poly(acrylamide) and ionic liquid. The modelling and section 2.2 helped a bit, but I think a comment on the interaction of lignin with the aforementioned components is necessary. Is lignin covalently attached to any of the components through the phenolic position for example? That is something that can be seen with already existing data of FT-IR or perhaps with a new HSQC/HMBC experiment.

Response Letter

Reviewer #1 (Remarks to the Author):

Guo and co-workers presented a photocuring resin made from lignin, acrylamide and 1-ethyl-3-methylimidazolium bromide, which showed enhanced room temperature phosphorescence (RTP) as well as longer emission lifetime. The resin was further employed to trigger a red afterglow emission when using rhodamine B as an energy transfer acceptor. The authors demonstrated the potential applications of this resin as luminescent coatings and inks for information encryption. Although it is an interesting study, the authors have extensively reported the RTP properties of lignin and its derivatives (Nat Comm 2022 and 2023; CRPS 2021), and lignin has been reported as a photoinitiator for 3D printing (ACS Sus Chem Eng, 2020, 8, 10959). As a result, the simple combination of these two properties, in my opinion, did not generate sufficient new knowledge that could match the high standard of Nature Communications. The detailed comments are as follows:

A: Thanks for the constructive comments.

Indeed, our group reported the preparation of a series of RTP materials from lignin (Nat Comm 2022 and 2023; CRPS 2021).

However, there are two main shortcomings in our previous research: 1) For these reports, lignin was processed in water solutions of H₂O₂, MgCl₂ or organic monomers. Although, the as-obtained materials can only generate RTP emission when they are dried using time-consuming natural evaporation or energy-consuming heating in an oven; 2) Additionally, integrating these materials with substrates to add function resulted in a decreased performance of the matrix since wetting or high-temperature thermal drying were involved. For example, when they are coated on cotton yarns or on a paper matrix assisted by water and thermal processing, the mechanical strength was decreased (**Supplementary Fig. 30, has been attached in response to this comment**).

As such these problems motivated us to look for a new generation of lignin-derived RTP materials. Compared to the energy-consuming oven technology for preparation, photocuring is an efficient and sustainable technology for producing materials. Also, photocuring technology is advantageous for integrating materials with functional substrates in an efficient and non-invasive manner, while no thermal or natural evaporation process is involved. Considering of these advantages, we developed a photocured RTP material from lignin.

Also, lignin has indeed been reported as a photoinitiator for a number of applications such as 3D printing. In particular, the Zhao, Baudis, and other groups have carried out excellent research in this area (ACS Sustain. Chem. Eng., 2019, 7, 4004; ACS Sustain. Chem. Eng., 2020, 8, 10959; Int. J. Biol. Macromol., 2022, 204, 234; Prog. Org. Coat., 2022, 173, 107210). Nevertheless, lignin was used as photoinitiator after chemical modification or in combination with co initiators. Due to the low activity of free radicals generated from untreated lignin, the potential of using lignin directly as photoinitiator in photopolymerization has rarely been

reported. Here we found that lignin in ionic liquids generated more reactive radicals than in water due to the special characteristics of the electrostatic interactions, hydrogen bonding interactions and viscosity of ionic liquids (European Polymer Journal, 2020, 133, 109778). As a result, lignin can initiate photopolymerization in an ionic liquid. (**Supplementary Fig. 8**).

Moreover, we found that lignin could not only serve as a photoinitiator and RTP chromophore, it can also enhance the mechanical performance of the cured samples, attributed to the interaction between lignin and the polymer matrix (Adv. Funct. Mater., 2019, 29, 1806912). Specifically, classic photoinitiator Ir-2959 was used as an alternative for lignin to initiate the same formula. The as-cured samples exhibited reduced hardness and Young's modulus, compared to the samples cured using lignin (**Supplementary Fig. 20, has been attached in answer to this comment**).

Based on these points, we have made conceptual advances in this research, when compared to these previous reports. However, we realize that we did not clarify these advances clearly in the previous version of our manuscript, as such we have now added these discussions in the revised manuscript to make the conceptual advances in the manuscript clear.

Supplementary Fig. 8 a) ESR spectra of lignosulfonate (0.1% w/w) in ionic liquid (blue line) or water (red line); b) Double bond conversion of P-Lig prepared in ionic liquid (blue line) or water (red line) (Inset: Digital images of the photocuring process of P-Lig in water).

1. In terms of the RTP coatings and inks for information encryption, what's the significant advantage of current design compared to other RPT systems? For example, RTP materials can be simply mixed with conventional photocuring resins. A photoinitiator may be needed, but the concentration is only around 1% or less.

A: Thanks for the comments.

In terms of the RTP coatings and inks for information encryption, what's the significant advantage of current design compared to other RPT systems?

To demonstrate the advantages, we compared cotton yarns and paper before and after

treatment with lignin-derived RTP materials prepared in the previous report (Cell Rep. Phys. Sci., 2021, 2, 100542). We observed that both the tensile stress and strain of the yarns (paper) decreased after the treatment with previously reported RTP materials (Supplementary Fig. 30). While, an obvious enhancement on mechanical performance was observed for the samples treated using the photocured RTP materials from this work, attributed to the robust mechanical performance of the photocured P-Lig. These results indicate that the previously reported RTP materials during preparation could result in weaker matrices and confirmed the advantages of the as-developed photocured lignin RTP materials.

RTP materials can be simply mixed with conventional photocuring resins. A photoinitiator may be needed, but the concentration is only around 1% or less.

Indeed, RTP chromophores could be mixed with conventional photocured resins with a classic photoinitiator to obtain photocured RTP materials. However, when we use lignin as RTP chromophore and added it into a photocured formula initiated by typical photoinitiator (0.1% Ir2959 or benzophenone). The formula was photocured with a double bond conversion of 99% and the as-obtained samples (Ir2959-P-Lig and BP-P-Lig) exhibited a much shorter RTP lifetime (Supplementary Fig. 19). Additionally, to demonstrate the quenching effect of the photoinitiator on the chromophore, phenylboronic acid as RTP chromophore was embedded into a photocured formula initiated by typical photoinitiator (Ir2959 or benzophenone 0.1%) and also exhibited a decreased lifetime when compared to the materials cured in the absence of Ir2959 or benzophenone (Supplementary Fig. 19). Thus, the optical properties of these RTP chromophores are affected by the added photoinitiator even when added in small amounts. Moreover, we found that the samples initiated by lignin exhibited higher mechanical performance including hardness and young's modulus than the photocured samples (Ir2959-P-AM) or with RTP chromophore (phenylboronic acid, Ir2959-P-BpA) initiated by typical photoinitiator (Ir2959) (Supplementary Fig. 20). It is important to note that the robust mechanical performance of RTP materials is beneficial for their application as coatings or functional components in devices. Specifically, Ir2959-P-AM was prepared using the same formula as P-Lig in the absence of lignin and initiated by Ir2959. Ir2959-P-BpA was also prepared using the same formula as P-Lig using phenylboronic acid as the RTP chromophore instead of lignin and initiated by Ir2959. Thus, using lignin as the photoinitiator and RTP chromophore at the same time could avoid the unexpected decreased RTP lifetime caused by interactions between the photoinitiator and RTP chromophore. Moreover, lignin can also enhance the mechanical performance of the photocured samples, which was beneficial for its practical applications.

Therefore, our research represents a significant conceptual advance when compared to the previous research.

Supplementary Fig. 30 a) Tensile strength and strain of original yarns (red line), yarns treated by lignosulfonate-derived RTP materials prepared in the previous reports (black line) and yarns treated by the photocured RTP materials in this work (blue line); b) Tensile strength and strain of original paper (red line), papers treated by lignosulfonate-derived RTP materials prepared in the previous reports (black line) and yarns treated by the photocured RTP materials in this work (blue line).

Supplementary Fig. 19 a) Phosphorescence lifetime of P-Lig (lignosulfonate as the chromophore) initiated by lignosulfonate (black), Ir2959 (red) and benzophenone (blue); b) Phosphorescence lifetime of P-BpA (phenylboronic acid as the chromophore) initiated by ammonium persulphate (black), Ir2959 (red) and benzophenone (blue). In all cases $\lambda_{exc.} = 320$ nm, $\lambda_{collected} = 510$ nm, delay time = 10 ms.

Supplementary Fig. 20 Nanoindentation tests of a) P-Lig, b) Ir2959-P-AM, c) Ir2959-P-BpA; Comparison of **d) Hardness and e) Young's modulus** of the mentioned above three materials.

2. The authors claimed that “the liquid surrounding of P-Lig contributes to RTP quenching” (line 92). Does it mean the free polymer chains of lignin in organic solvents make the excited state go back to ground state with more nonradiative relaxations, or the acrylamide has charge transfer interactions with lignin that can quench the emission?

A: Thanks for the comments. To understand the reason for the RTP quenching, we dissolved the lignin in the ionic liquid without acrylamide. We found that no RTP emission was observed. Thus, the free polymer chains of lignin in organic solvents, result in the excited states returning to the ground states due to more nonradiative relaxation facilitated by the solution state, and contributes to the quenching of RTP.

We added the results as **Supplementary Fig. 11**

Supplementary Fig. 11 a) Fluorescence and phosphorescence emission spectra of lignosulfonate dissolved in ionic liquid; b) Phosphorescent lifetime of the lignosulfonate dissolved in ionic liquid (no signal detected). In all cases $\lambda_{exc.} = 320$ nm, $\lambda_{collected} = 510$ nm, delay time = 10 ms.

3. Line 104, how to understand “Since the polyacrylamide possesses a low compatibility with these solvents, the solvents are unable to quench the RTP emissions”? Did the authors investigate the relationships between compatibility and RTP intensity?

A: Thanks for the comments. We investigated the relationship between compatibility and RTP intensity. Polyacrylamide possesses a high compatibility with water and low compatibility with organic solvents. Thus, water/humidity quenched the RTP of P-Lig while organic solvents could not. This result indicated that RTP emission of P-Lig was easily quenched when it is treated with a compatible solvents. Additionally, we also systematically investigated the relationship between compatibility and RTP of the sample in water. The RTP lifetime decreased continuously after immersing it in water for increasing times. After 30 mins, it only exhibited a lifetime of 13.74 ms and after 40 mins, no RTP emission was observed.

We added the discussion in the revised manuscript and the result as **Supplementary Fig. 15**.

Supplementary Fig. 15 Phosphorescence lifetime of P-Lig after immersion in water for different times.

4. What's the factor leading to short-lived RTP emission of the formula without lignin?

A: Thanks for the comments. This is because the carbonyl moieties in poly-acrylamide formed clusters and through-space conjugation occurred, which triggered the short-lived emission. (J. Am. Chem. Soc., 2021 143, 16256; Adv. Mater., 2023, 35, 2300244.)

We added the explanation and the references in the revised manuscript.

5. The lignin-acrylamide interaction model for theoretical calculations is based on one single lignin group and one acrylamide monomer? The proper comparison should be based on same amount of lignin groups as well as the monomeric units. For example, if the interaction between lignin and a 6-unit polyacrylamide oligomer is investigated, the comparable model for monomer should be lignin and 6 monomer molecules. The reality in photocuring resins is that numerous monomers exist surrounding the lignin.

A: Thanks for the comments. We calculated the interaction between lignin and 6 monomer molecules. The value was -13.82 eV.

We added the results as **Supplementary Fig. 27**.

Supplementary Fig. 27 Calculated interaction model between lignosulfonate and different number of acrylamide monomers (Left means lignin with two monomers, the value was -10.39 eV and right means lignin with six monomers, the value was -13.82 eV).

6. What if other monomers are used in this system, such as acrylates, methacrylates, and styrenics? Does this strategy have potential broad applications in different occasions? A more systematic study with deep understanding would be more convincing.

A: Thanks for the comments. We checked the potential for the broader application using different formulas. The formula can work well when we introduced monomers, such as, acrylic acid, methyl acrylate, methyl methacrylate, styrene and N-isopropylacrylamide, into the mixture of ionic liquid and lignin. All the samples can be photocured in 20 mins and exhibited RTP emission with lifetimes of 76.96, 62.95, 57.95, 67.19 and 85.56 ms.

We added the results as **Supplementary Fig. 21**.

Supplementary Fig. 21 Fluorescence and phosphorescence emission spectra of a) P-AA ($\lambda_{exc.} = 310$ nm, $\lambda_{collected} = 510$ nm), b) P-MA ($\lambda_{exc.} = 320$ nm, $\lambda_{collected} = 510$ nm), c) P-MMA ($\lambda_{exc.} = 320$ nm, $\lambda_{collected} = 510$ nm), d) P-SM ($\lambda_{exc.} = 330$ nm, $\lambda_{collected} = 500$ nm), e) P-NIPAM ($\lambda_{exc.} = 310$ nm, $\lambda_{collected} = 490$ nm), Inset: the images of polymers in daylight (left), polymers upon excitation by UV light source (middle) and polymers after switching off the UV light source (right); f) Phosphorescence lifetime of the polymers mentioned above at the corresponding emission wavelength. In all cases the delay time is 10 ms.

7. The FTIR data may be provided for evaluating the double bond conversion.

A: Thanks for the comments. We added all the FT-IR data for the samples.

FTIR spectra a) - e) FT-IR spectra of the photocured materials using different monomers in this system upon UV irradiation for different times (acrylic acid, methyl acrylate, methyl methacrylate, styrene and N-isopropylacrylamide); f) Conversion of double bonds of the materials mentioned above upon UV irradiation for different times; g) – i) FT-IR spectra of the photocured materials using different ionic liquids in this system upon UV irradiation for different times (1-ethyl-3-methylimidazolium bromide, 1-ethyl-3-methylimidazolium chloride, 1-ethyl-3-methylimidazolium acetate); j) FT-IR spectra of P-Lig that initiated by Ir2959 (Ir2959-P-Lig); k) FT-IR spectra of P-Lig that initiated by benzophenone (BP-P-Lig); l) Double bond conversion of Ir2959-P-Lig and BP-P-Lig upon UV irradiation for different times.

Reviewer #2 (Remarks to the Author):

In the manuscript entitled “Title: Photocured room temperature phosphorescence materials from lignin” authors have developed a photocured phosphorescent material (P-Lig) using lignin, acrylamide and an ionic liquid. Authors have used the same material (lignin) that can act both as photoinitiator to initiate the polymerization of acrylamide upon UV irradiation, and as phosphor that is trapped inside the polymer network. Authors have also shown a red-afterglow by using rhodamine B as the acceptor in the energy-transfer process along with P-Lig. Finally, authors have also shown different applications using the same material as photo-curable coatings and inks for information encryption. Same author have shown phosphorescence using lignin (Nat. Commun. 13, 5508 (2022); Nat. Commun. 14, 2614 (2023)) in previous reports. Also, there are reports on photocurable phosphorescence (Adv. Optical Mater. 2300240 (2023)). Authors should elaborate more on the novelty aspects in the revised manuscript apart from addressing the following concerns from my end.

A: Thanks for the constructive comments.

Indeed, our group reported the preparation of a series of RTP materials from lignin (Nat Comm 2022 and 2023; CRPS 2021).

However, there are two main shortcomings in our previous research: 1) For these reports, lignin was processed in water solutions of H₂O₂, MgCl₂ or organic monomers. Although, the as-obtained materials can only generate RTP emission when they are dried using time-consuming natural evaporation or energy-consuming heating in an oven; 2) Additionally, integrating these materials with substrates to add function resulted in a decreased performance of the matrix since wetting or high-temperature thermal drying were involved. For example, when they are coated on cotton yarns or on a paper matrix assisted by water and thermal processing, the mechanical strength was decreased (**Supplementary Fig. 30**).

As such these problems motivated us to look for a new generation of lignin-derived RTP materials. Compared to the energy-consuming oven technology for preparation, photocuring is an efficient and sustainable technology for producing materials. Also, photocuring technology is advantageous for integrating materials with functional substrates in an efficient and non-invasive manner, while no thermal or natural evaporation process is involved. Considering of these advantages, we developed a photocured RTP material from lignin.

Also, photocurable phosphorescence (Adv. Optical Mater. 2300240 (2023)) was indeed reported and we also cited this reference in our manuscript. In the work, two different materials including petrol derived diphenyl (2,4,6-trimethylbenzoyl) phosphine oxide (TPO) and 2,3-naphthalenedicarboxylic anhydride were used as the photosensitizer and RTP chromophore, respectively. While, sustainable lignin simultaneously serves as photosensitizer and RTP chromophores in our work, which is much more convenient. Additionally, the performance of RTP chromophores can be negatively affected by an external photoinitiator in photocured RTP materials. For example, when lignin is used as a RTP chromophore and added it into a photocured formula initiated by typical photoinitiator (Ir2959 or benzophenone, 0.1%).

The obtained samples exhibited a much shorter RTP lifetime (Supplementary Fig. 19). In addition to demonstrate the quenching effect of the photoinitiator on the chromophore, phenylboronic acid was embedded as RTP chromophore into a photocured formula initiated by a typical photoinitiator (Ir2959 or benzophenone, 0.1%) and exhibited a decreased lifetime when compared with the ammonium persulfate-cured materials. (Supplementary Fig. 19). Thus, the optical properties of RTP chromophores are affected by the additional photoinitiator even when present at very small amounts.

Moreover, we found that the cured samples initiated by lignin exhibited higher mechanical performance including hardness and young's modulus than the photocured samples (Ir2959-P-AM and Ir2959-P-BpA) without or with RTP chromophore (phenylboronic acid) initiated by typical photoinitiator (Ir2959) (Supplementary Fig. 20). It is important to note that the robust mechanical performance of RTP materials is beneficial for their application as coatings or functional components in devices.

Specifically, Ir2959-P-AM was prepared using the same formula as P-Lig in the absence of lignin and initiated by Ir2959. Ir2959-P-BpA was also prepared using the same formula as P-Lig while using phenylboronic acid as the RTP chromophore instead of lignin and initiated by Ir2959.

As such we think this research represents a significant conceptual advance, compared to previous research.

Supplementary Fig. 30 a) Tensile strength and strain of original yarns (red line), yarns treated by lignosulfonate-derived RTP materials prepared in the previous reports (black line) and yarns treated by the photocured RTP materials in this work (blue line); b) Tensile strength and strain of original paper (red line), papers treated by lignosulfonate-derived RTP materials prepared in the previous reports (black line) and yarns treated by the photocured RTP materials in this work (blue line).

Supplementary Fig. 19 a) Phosphorescence lifetime of P-Lig (lignosulfonate as the chromophore) initiated by lignosulfonate (black), Ir2959 (red) and benzophenone (blue); b) Phosphorescence lifetime of P-BpA (phenylboronic acid as the chromophore) initiated by ammonium persulphate (black), Ir2959 (red) and benzophenone (blue). In all cases $\lambda_{exc.} = 320$ nm, $\lambda_{collected} = 510$ nm, delay time = 10 ms.

Supplementary Fig. 20 Nanoindentation tests of a) P-Lig, b) Ir2959-P-AM, c) Ir2959-P-BpA; Comparison of d) **Hardness** and e) **Young's modulus** of the mentioned above three materials.

1. Authors should mention the delay time used for delayed emission spectrum in Figure 2d.

A: Thanks for the comments. The delay time was 10 ms. We added this information to the revised legend of Figure 2d.

2. In line no. 97, authors have mentioned the decrement of lifetime to 24 ms on humidity treatment. However, the lifetime plot for the same is not shown.

A: Thanks for the comments. We added the lifetime plot. We added the results as **Supplementary Fig. 14**.

Supplementary Fig. 14 a) RTP lifetime of P-Lig upon recycling of humidity (80 %) and drying (80 °C) cycles; b) Phosphorescent lifetime of P-Lig upon humidity treatment.

3. What is the phosphorescent quantum yields for these materials.

A: Thanks for the comments. The phosphorescence quantum yield of P-Lig is 11.04% and the fluorescence and delayed fluorescence quantum yield of P-Lig/RhB is 57.62%. We added this information to the revised manuscript.

4. What is the type of energy-transfer happening in this case. FRET or TS-FRET. Does the triplet state of lignin is alone involved in the process. What is the efficiency of the energy-transfer process here. It is recommended to investigate further on the energy-transfer process happening here since the manuscript lacks convincing energy transfer data from P-Lig to RhB. To address this gap, provide gradual donor (P-Lig) quenching data, including donor emission intensity and corresponding lifetime, with varying concentrations of RhB in the system. Additionally, demonstrate whether, upon indirect excitation (exciting the donor), the emission intensity of RhB surpasses that of direct excitation. This analysis will provide crucial insights into the energy transfer dynamics between P-Lig and RhB.

A: Thanks for the comments. As the referee suggested, we measured gradual donor (P-Lig) quenching data, including donor emission intensity and corresponding lifetime, with varying concentrations of RhB in the system (**Supplementary Fig. 24**). Specifically, the emission spectra of P-Lig/RhB ($\lambda_{exc.} = 320$ nm) exhibits a gradual decrease of the lignin phosphorescence

emission centered at 510 nm and a concomitant enhancement of RhB fluorescence in the 560–700 nm region with increasing the concentrations of RhB in this system from 0 to 1000 ppm, which indicates an efficient energy transfer from the triplet state of the donor to the acceptor molecules. Furthermore, time-resolved emission lifetime analyses of the lignin phosphorescence monitored at 510 nm ($\lambda_{\text{exc.}} = 320$ nm), exhibits a gradual decrease of the average lifetime from 110.32 ms to 22.34 ms with loading of RhB. Additionally, we observed that upon indirect excitation at 320 nm (exciting the donor), the emission intensity of RhB surpasses that of direct excitation (**Supplementary Fig. 24**). This indicates that the long-lived triplet excitons of the donor lignin molecules are the source of the delayed population of the singlet state of the acceptor RhB via an efficient triplet-to-singlet FRET. Also, the decreased lifetime of donor with increased loading amount of RhB clearly eliminate the possibility of simple energy transfer (emission–reabsorption process) between lignin and RhB. This is because the donor lifetime is not expected to change for this situation.

All these results confirmed that triplet-to-singlet FRET occurred between lignin and RhB in the photocured matrix. The efficiency was 16.5%, 19.4%, 23.9%, 34.0%, 45.9%, 64.6% and 79.7% with an increasing concentration of RhB (**Table S1**). (Angew. Chem. Int. Ed., 2020, 59, 9393; Angew. Chem. Int. Ed., 2021, 60, 19720; Adv. Funct. Mater., 2020, 30, 2003693.)

Supplementary Fig. 24 a) Delayed emission spectra of P-Lig/RhB ($\lambda_{exc.} = 320$ nm); b) Lifetime decay plots of P-Lig/RhB ($\lambda_{collected} = 510$ nm) and c) Lifetime decay plots of P-Lig/RhB ($\lambda_{collected} = 600$ nm) with increased doping concentration of RhB; d) Delayed emission spectra of P-Lig/RhB (100:0.07) upon direct excitation ($\lambda_{exc.} = 550$ nm, black line) and upon indirect excitation ($\lambda_{exc.} = 320$ nm, red line), the delay time was 10 ms.

Table S1 Summary of energy transfer (Φ_{et}) efficiency.

Acceptor	Donor(P-Lig) and Acceptor (RhB) Ratio (w/w)	Average lifetime (in ms) of P-Lig at 510 nm ($\lambda_{exc.} = 320$ nm)	Energy Transfer Efficiency (%)
RhB	100:0	110.32	-
RhB	100:0.01	92.09	16.5
RhB	100:0.03	88.89	19.4
RhB	100:0.05	83.91	23.9
RhB	100:0.07	72.78	34.0
RhB	100:0.08	59.67	45.9
RhB	100:0.09	39.08	64.6
RhB	100:0.10	22.34	79.7

5. It is advisable to clearly mention the excitation and collection wavelengths in the figure captions throughout the manuscript, including both the main text and supplementary information for better understanding for the readers.

A: Thanks for the comments. We added the excitation and collection wavelengths in the figure captions throughout the manuscript.

6. In line no. 131, authors have claimed that the “as-prepared sample only exhibited weak and short-lived RTP emissions”. Please provide the corresponding steady-state emission spectra. Additionally, clarify whether this short-lived RTP emission is observed in the absence of lignin, and if so, provide insights into the source of this RTP emission.

A: Thanks for the comments.

We added the corresponding steady-state emission spectra (**Supplementary Fig. 26**).

The carbonyl moieties in the as-generated polyacrylamide formed molecular clusters and through-space conjugation occurred, which triggered the short-lived emission. (J. Am. Chem. Soc., 2021 143, 16256; Adv. Mater., 2023, 35, 2300244.)

Supplementary Fig. 26 Fluorescence and phosphorescence emission spectra of P-AM initiated by ammonium persulfate ($\lambda_{exc.} = 320$ nm, $\lambda_{collected} = 510$ nm, delay time = 10 ms).

7. Authors have achieved afterglow, and also utilized energy-transfer process to sensitize the delayed fluorescence from rhodamine B. In this context, the following papers should be cited: *Angew. Chem. Int. Ed.* **60**, 19720–19724 (2021); *Adv. Funct. Mater.* **30**, 2003693 (2020).

A: Thanks for the comments. We have added all these references. Additionally, we also cited *Angew. Chem. Int. Ed.*, 2020, 59, 9393 in the manuscript.

Reviewer #3 (Remarks to the Author):

I would like to start by stating that the manuscript titled: “Photocured room temperature phosphorescence materials from lignin” reports a novel, easy-to-follow and “simple” approach to prepare RTP materials based on lignosulfonate. I think that this manuscript should be accepted for publication, however, I think that some modifications are necessary. I have divided my comments in different sections so it is easier for the authors to address.

A: Thanks for the constructive comments.

Title:

The title of the manuscript gives the impression that it refers to a review article rather than a paper. I suggest the authors to change it and also replace the word “lignin” with “lignosulfonate”

A: Thanks for the comments, we revised it according to your suggestion.

Abstract, Introduction and Conclusions:

The abstract and conclusions sections contain the necessary information for a reader to get an overview of the paper and its most important findings. They are also well-written!

As for the introduction it is short and concise.

However, I suggest the word lignin to be replaced with lignosulfonate in the entire text as there are many lignin types that could fall into this category.

A: Thanks for the comments, we revised it according to your suggestion.

Results and Discussion:

• Why did the authors used lignosulfonates compared to other more industrially relevant technical lignins such as kraft? I think this should be included in the main manuscript.

Would the results be the same is the experiments were to be repeated with kraft lignin instead of lignosulfonate?

A: Thanks for the comments, we have used kraft lignin, alkali lignin and enzymatic hydrolysis lignin and found that all of them can serve as photoinitiator for the reaction. All the samples were photocured in 20 mins with double bond conversion of 89.27%, 97.24% and 73.89%, respectively. Additionally, all of these cured samples generate RTP emissions.

We added the results as **Supplementary Fig. 22**.

Supplementary Fig. 22 Fluorescence and phosphorescence emission spectra of a) P-Kraft (Kraft lignin), b) P-EL (Enzymatic hydrolysis lignin), c) P-AL (alkali lignin), Inset: The images of polymers in daylight (left), polymers upon excitation by UV light source (middle) and polymers after switching off the UV light source (right); d) Phosphorescence lifetime of the polymers mentioned above. In all cases $\lambda_{exc.} = 320$ nm, $\lambda_{collected} = 510$ nm, delay time = 10 ms.

• Has the same behaviour, i.e., lignosulfonate act as radical generator, be reported elsewhere in the literature with a similar ionic liquid system? If yes, I think it should be mentioned somewhere in lines 79-81.

A: Thanks for the comments, lignin has been used as a photoinitiator for lots of applications after suitable chemical modification or assisted by co initiators (ACS Sustain. Chem. Eng., 2019, 7, 4004; ACS Sustain. Chem. Eng., 2020, 8, 10959; Int. J. Biol. Macromol., 2022, 204, 234; Prog. Org. Coat., 2022, 173, 107210). Nevertheless, because of the low activity of the free radicals generated from untreated lignin, exploration of the potential of lignin as a photoinitiator in photopolymerization, has rarely been reported.

Additionally, we found that lignosulfonate in ionic liquids can generate more photoradicals, compared to water. As a result, lignosulfonate in ionic liquids successfully initiated the polymerization. We added the results as **Supplementary Fig. 8**. We also added this discussion and cited new references in the revised manuscript.

Supplementary Fig. 8 a) ESR spectra of lignosulfonate (0.1% w/w) in ionic liquid (blue line) or water (red line); b) Double bond conversion of P-Lig prepared in ionic liquid (blue line) or water (red line) (Inset: digital images of the photocuring process of P-Lig in water).

• It is well-known that lignin act as a radical scavenger due to its phenolic content. Can the authors comment on this and how can they address the balance between radical generation and radical scavenging?

A: Thanks for the comments. The referee is right and the phenolic moieties in lignin would quench the generated radicals. Theoretically, concentrated lignin would generate more photoradicals. However, concentrated lignin results in more phenolic moieties, which reduces the amount of photoradicals. Thus, it is important to use a suitable concentration of lignin that can generate the maximum amount of photoradicals. We used the ESR spectra to determine the best concentration range of lignosulfonate for generating photoradicals 0.1-0.15% w/w in the ionic liquid. Thus, we can optimize the concentration of lignin to address the balance between radical generation and radical scavenging.

We added the result as **Supplementary Fig. 4**.

Supplementary Fig. 4 ESR spectra of lignosulfonate with different concentration in ionic liquid upon UV irradiation for 10 min.

• Can the authors comment on how the temperature would affect the performance of the lignin-containing RTP materials?

A: Thanks for the comments. We measured the RTP emission and lifetime of P-Lig at different temperatures. An increased temperature promoted the non-radiative migration of the excitons of lignin, which compromised both the RTP intensity and lifetime.

We added the result as **Supplementary Fig. 18**.

Supplementary Fig. 18 Temperature dependent a) Phosphorescence spectra and d) Lifetime of P-Lig ($\lambda_{exc.} = 320$ nm, $\lambda_{collected} = 510$ nm).

Methods:

Can the authors comment on the reproducibility of the method they developed?

A: Thanks for the comments. We reproduced the materials using our method 5 times and the

as-obtained materials exhibited good reproducibility with a stable lifetime of 110 ms.

We added the result as Supplementary Fig. 13.

Supplementary Fig. 13 Phosphorescent lifetime of P-Lig produced using the same method 5 times. In all cases $\lambda_{\text{exc.}} = 320$ nm, $\lambda_{\text{collected}} = 510$ nm, delay time = 10 ms.

Also, was there any optimization performed for the quantities used in this formulation? Could perhaps a higher liginosulfonate content reduce the UV irradiation time?

A: Thanks for the comments. According to the ESR result we performed, we increased the concentration of liginosulfonate from 0.1% to 0.15% w/w, which is the most suitable concentration range for generating photoradicals. We found that higher liginosulfonate content indeed reduced the UV irradiation time. We added the result as Supplementary Fig. 6.

Supplementary Fig. 6 Double bond conversion of P-Lig prepared using different concentrations of liginosulfonate.

General questions:

Can the authors comment on the recyclability of the ionic liquid used in their protocol? I think that this is quite important and it should be mentioned in the main manuscript too.

A: Thanks for the comments. The ionic liquid was recyclable. Specifically, the P-Lig was dissolved in the water and was treated using a dialysis bag for 5 h. The as-obtained water solution outside the dialysis bag was washed using ethyl acetate. Then, the recycled ionic liquid was obtained by evaporating the water. For the whole process, ~96% of the ionic liquid was recycled. ^1H NMR analysis indicated that the recycled ionic liquid exhibited the same signals as the virgin ionic liquid. Moreover, P-Lig was prepared using recycled ionic liquid and exhibited RTP lifetimes of 110.09 ms, which is similar to the initial P-Lig samples. All these experiments confirmed that ionic liquid could be efficiently recycled from P-Lig and reused. We added these results in the main manuscript and also added the data as Supplementary Fig. 31 and 32.

Supplementary Fig. 31 ^1H NMR spectrum of a) 1-ethyl-3-methylimidazolium bromide purchased from Aladdin (Shanghai, China), and b) Recycled ionic liquid form P-Lig.

Supplementary Fig. 32 a) Standard and delayed emission spectra of P-Lig prepared using recycled ionic liquid; b) Phosphorescence lifetime of the P-Lig mentioned above ($\lambda_{\text{exc.}} = 320 \text{ nm}$, $\lambda_{\text{collected}} = 510 \text{ nm}$, delay time = 10 ms).

Based on the explanation in the main text of the manuscript as well as from the methods section, it is my understanding that the lignosulfonate content is kept as low as possible. However, since no purification is performed after the polymerization is complete, I am curious about the interaction of lignin with the rest of the poly(acrylamide) and ionic liquid. The modelling and section 2.2 helped a bit, but I think a comment on the interaction of lignin with the aforementioned components is necessary. Is lignin covalently attached to any of the components through the phenolic position for example? That is something that can be seen with already existing data of FT-IR or perhaps with a new HSQC/HMBC experiment.

A: Thanks for the comments. To understand the possible linkage between lignin and the as-formed polymers in P-Lig, the ionic liquid and unreacted monomers were separated from the P-Lig using dialysis. The solid was then characterized by ^1H NMR. The signals of phenolic moieties in the lignosulfonate disappeared. While, new signals, assigned as $-\text{CH}_2-$ formed between the phenolic position of lignosulfonate and the polymer, were observed (ChemSusChem., 2021, 14, 1184; Eur. Polym. J., 2015, 70, 371). As a control, the signals of phenolic moieties of lignosulfonate were still observed in the physical mixture of lignosulfonate and acrylamide without light irradiation. Also, the mixture did not exhibit signals for $-\text{CH}_2-$ formed between the phenolic position of lignosulfonate and the polymer. Accordingly, enhanced signals for C-O-C were observed for the in situ FT-IR spectra. All these results suggest that lignin was covalently attached to the polymer through the phenolic positions. We added the results as **Supplementary Fig. 9**

Supplementary Fig. 9 ^1H NMR spectrum of a) lignosulfonate, b) the products of the reaction between lignosulfonate and acrylamide in ionic liquid upon UV irradiation, and c) physical mixture of lignosulfonate and acrylamide without light irradiation; d) In situ FT-IR spectra of P-Lig upon UV irradiation for different times.

Reviewers' Comments:

Reviewer #1:

Remarks to the Author:

The authors have mostly addressed my concerns regarding the novelty and the experimental results. To my understanding, the introduction of lignin can prevent the RTP quenching when using conventional photoinitiators, is a key point of this work. This has not been reflected in the introduction section. The authors may emphasize this point for better clarification.

In addition, to show the enhancements of mechanical properties, average value+standard deviation may be used if triplicates or more are tested (e.g., Fig S20) for higher reliability. This applies also to lifetime (e.g., Fig. 3C).

Reviewer #2:

Remarks to the Author:

Authors have addressed all my concern's during the revision and I am very happy to recommend this contribution for publications.

Reviewer #3:

Remarks to the Author:

In my opinion, the authors address all of my comments in a satisfactory way. Hence, I propose for their manuscript to be accepted without any further modification.

Response

Reviewer #1 (Remarks to the Author):

The authors have mostly addressed my concerns regarding the novelty and the experimental results. To my understanding, the introduction of lignin can prevent the RTP quenching when using conventional photoinitiators, is a key point of this work. This has not been reflected in the introduction section. The authors may emphasize this point for better clarification.

A: Thanks for the comments. We emphasize this point in the introduction.

“Moreover, the photosensitizer could also quench the RTP emission from the chromophores. Additionally, lignin is the most abundant aromatic biomass resource in nature and exhibits interesting optical properties, such as UV blocking, fluorescence, photothermal conversion, RTP and photocatalytic properties. In this work, we employ liginosulfonate to serve as RTP chromophore and photoinitiator simultaneously. Specifically, liginosulfonate generates radicals to polymerize the acrylamide in the ionic liquid (1-ethyl-3-methylimidazolium bromide) upon UV irradiation. As a result, the liginosulfonate is confined in the as-formed bulk and transparent matrix (P-Lig) generating a material able to generate RTP from the confined liginosulfonate in P-Lig. Significantly, this design can efficiently prevent the quenching of RTP that occurs when using a conventional formula consisting of standard photoinitiators and RTP chromophores.”

In addition, to show the enhancements of mechanical properties, average value+standard deviation may be used if triplicates or more are tested (e.g., Fig S20) for higher reliability. This applies also to lifetime (e.g., Fig. 3C).

A: Thanks for the comments. We added the error bar in these Figures.

Figure 3. Mechanism for the RTP of P-Lig. a) In situ FT-IR spectra of P-Lig upon

UV irradiation for 0 s (black line), 150 s (red line) and 600 s (blue line); b) Calculated interaction model between lignosulfonate and acrylamide with different polymerization degrees; c) Lifetime of P-Lig prepared from ionic liquid with different anions (Error bars indicate the standard deviation for three separate measurements of the samples, the lifetime of the sample containing Br^- is 109.72 ± 0.52 ms, the lifetime of the sample containing Cl^- is 89.27 ± 0.43 ms, the lifetime of the sample containing CH_3COO^- is 38.67 ± 0.41 ms).

Supplementary Fig. 20 Mechanical properties of P-Lig, Ir2959-P-AM and Ir2959-P-BpA. Nanoindentation tests of a) P-Lig, b) Ir2959-P-AM, c) Ir2959-P-BpA; Comparison of d) Hardness and e) Young's modulus of the mentioned above three materials (Error bars indicate the standard deviations for three separate measurements of the samples, the value for hardness and young's modulus of P-Lig are 112.33 ± 0.237 Mpa and 3.567 ± 0.015 Gpa, respectively; the value for hardness and young's modulus of Ir2959-P-AM are 30.86 ± 0.726 Mpa and 1.44 ± 0.026 Gpa, respectively; the value for hardness and young's modulus of Ir2959-P-BpA are 32.25 ± 1.041 Mpa and 1.49 ± 0.047 Gpa, respectively).

Reviewer #2 (Remarks to the Author):

Authors have addressed all my concern's during the revision and I am very happy to recommend this contribution for publications.

A: Thanks for the comments.

Reviewer #3 (Remarks to the Author):

In my opinion, the authors address all of my comments in a satisfactory way. Hence, I propose for their manuscript to be accepted without any further modification.

A: Thanks for the comments.